# Syndecan-1 Plays a Role in the Pathogenesis of Sjögren’s Disease by Inducing B-Cell Chemotaxis through CXCL13–Heparan Sulfate Interaction

**DOI:** 10.3390/ijms25179375

**Published:** 2024-08-29

**Authors:** Nan Young Lee, Hirut Yadeta Ture, Eun Ju Lee, Ji Ae Jang, Gunwoo Kim, Eon Jeong Nam

**Affiliations:** 1Department of Clinical Pathology, School of Medicine, Kyungpook National University, Daegu 41405, Republic of Korea; leenanyoung70@gmail.com; 2Division of Rheumatology, Department of Internal Medicine, School of Medicine, Kyungpook National University, Daegu 41405, Republic of Korea; danucharmy@gmail.com; 3Laboratory for Arthritis and Bone Biology, Fatima Research Institute, Daegu Fatima Hospital, Daegu 41199, Republic of Korea; cord2007@hanmail.net (E.J.L.); wkdwldo14@naver.com (J.A.J.); drgunwoo@gmail.com (G.K.)

**Keywords:** Sjögren’s disease, syndecan-1, epithelial cell, CXCL13, B cell

## Abstract

In Sjögren’s disease (SjD), the salivary glandular epithelial cells can induce the chemotaxis of B cells by secreting B-cell chemokines such as C-X-C motif chemokine ligand 13 (CXCL13). Syndecan-1 (SDC-1) is a major transmembrane heparan sulfate proteoglycan (HSPG) predominantly expressed on epithelial cells that binds to and regulates heparan sulfate (HS)-binding molecules, including chemokines. We aimed to determine whether SDC-1 plays a role in the pathogenesis of SjD by acting on the binding of HS to B-cell chemokines. To assess changes in glandular inflammation and SDC-1 concentrations in the submandibular gland (SMG) and blood, female NOD/ShiLtJ and sex- and age-matched C57BL/10 mice were used. In the SMG of NOD/ShiLtJ mice, inflammatory responses were identified at 8 weeks of age, but increased SDC-1 concentrations in the SMG and blood were observed at 6 weeks of age, when inflammation had not yet started. As the inflammation of the SMG worsened, the SDC-1 concentrations in the SMG and blood increased. The expression of the CXCL13 and its receptor C-X-C chemokine receptor type 5 (CXCR5) began to increase in the SMG at 6 weeks of age and continued until 12 weeks of age. Immunofluorescence staining in SMG tissue and normal murine mammary gland cells confirmed the co-localization of SDC-1 and CXCL13, and SDC-1 formed a complex with CXCL13 in an immunoprecipitation assay. Furthermore, NOD/ShiLtJ mice were treated with 5 mg/kg HS intraperitoneally thrice per week for 6–10 weeks of age, and the therapeutic effects in the SMG were assessed at the end of 10 weeks of age. NOD/ShiLtJ mice treated with HS showed attenuated salivary gland inflammation with reduced B-cell infiltration, germinal center formation and CXCR5 expression. These findings suggest that SDC-1 plays a pivotal role in the pathogenesis of SjD by binding to CXCL13 through the HS chain.

## 1. Introduction

Sjögren’s disease (SjD) is a prototypical autoimmune disease characterized by chronic lymphocytic infiltration in the exocrine glands, preferentially affecting the salivary and lacrimal glands, although other organs can also be involved [1,2,3]. The complex interplay between salivary gland epithelial cells (SGECs) and innate and adaptive immunity is critical in the pathogenesis of SjD. B cells play a central role in this process, becoming overactivated and functioning as cytokine-secreting, antigen-presenting cells or autoantibody producers [1,2]. C-X-C motif chemokine ligand 13 (CXCL13) is a strong chemokine for B cells and is produced mainly by mesenchymal lymphoid tissue organizer cells, follicular dendritic cells, and human T-follicular helper cells [4,5], in addition to non-immune cells resident in salivary glands, including ductal epithelial cells and endothelial cells in the salivary glands of SjD [6]. By binding to C-X-C chemokine receptor type 5 (CXCR5), CXCL13 plays an important role in lymphoid neogenesis, lymphoid organization, and immune responses [5] and is associated with germinal center (GC)-like structures in patients with SjD [7].

SGECs are also major players in SjD pathogenesis as targets of disease and drivers of the disease process that promote the overactivation of the immune system [2,4]. The dysregulation of innate immune signaling pathways and consequent pro-inflammatory cytokine and chemokine production by SGECs leads to salivary gland inflammation and dysfunction [2,4].

Syndecan-1 (SDC-1) is the prototype cell surface heparan sulfate proteoglycan (HSPG) [8] and is predominantly expressed on epithelial and plasma cells in adult tissues [9,10,11]. It consists of an extracellular domain where heparan sulfate (HS) chains attach distally to the plasma membrane, followed by the signature transmembrane and cytoplasmic domains, which are highly homologous among various SDCs and species [9,10,11]. On the cell surface, SDC-1 binds to and regulates HS-binding molecules, including chemokines, cytokines, growth factors, and extracellular matrix components, thereby allowing SDC-1 to modulate multiple inflammatory responses by functioning as both pro- and anti-inflammatory molecules [9,10,11]. SDC-1 can modulate several chemokine functions by inducing oligomerization, which can potentiate chemokine activities, protecting chemokines from degradation and serving as a direct internalization receptor for specific chemokines [10,12]. However, specific binding partners of SDC-1 for chemokines, including CXCL13, have not been completely identified. SDC-1 also functions as a soluble HSPG in its extracellular domain, replete with all its HS chains, and can be proteolytically released from the cell surface through ectodomain shedding [9,10]. Shed HSPG ectodomains can also bind to and block chemokine functions [13].

Previously, we demonstrated that SDC-1 expression increased on SGECs in inflamed minor salivary glands of patients with SjD, and that the SDC-1 concentrations of saliva and blood were higher in patients with SjD than in healthy participants [14]. Additionally, the SDC-1 concentrations of saliva and blood reflect the salivary glandular function and disease activity, respectively. However, the role of SDC-1 in the pathogenesis of SjD has not been studied. In this study, we aimed to elucidate the role of SDC-1 in the pathogenesis of SjD by examining the expression of SDC-1 in submandibular salivary glands (SMGs) in NOD/ShiLtJ mice, a mouse model of SjD; the binding of CXCL13 to SDC-1; and the effects of HS treatment on SMG inflammation and B-cell infiltration.

## 2. Results

### 2.1. Histopathological Changes in the Salivary Glands of NOD/ShiLtJ Mice

Initially, we examined the severity of inflammation in the SMGs of NOD/ShiLtJ mice. Inflammatory cell infiltration was not observed in the salivary gland tissues of 6-week-old mice; however, it tended to increase with age (Figure 1). The mean ratio indexes were 0.0, 0.7, 10.0, and 15.7 in the 6-, 8-, 10-, and 12-week-old NOD/ShiLtJ mice, respectively (Figure 1B), and no significant difference was observed in salivary gland inflammation between 6- and 8-week-old mice (*p* = 0.25). However, significant differences in the severity of inflammation were observed between 8- and 10-week-old mice (*p* < 0.001) and between 10- and 12-week-old mice (*p* = 0.027). On examination of the incidence rate according to the degree of ratio index in mice of different ages, an inflammatory change was observed in all salivary gland tissues of 10- and 12-week-old mice and was more severe in 12-week-old mice than in 10-week-old mice (Figure 1C).

### 2.2. Increased SDC-1 Expression in Blood and SMGs of NOD/ShiLtJ Mice

To determine the changes in SDC-1 expression according to the degree of inflammation in SMGs, we identified SDC-1 expression in salivary gland tissues by immunofluorescence (IF) staining and measured the concentration of SDC-1 in SMGs and blood using the dot-blotting method. The expression of SDC-1 in IF-stained tissues increased as tissue inflammation worsened and was mainly localized to the epithelial cells lining the salivary duct (Figure 2A).

The SDC-1 concentrations in the blood (Figure 2B) and SMGs (Figure 2C) significantly increased in NOD/ShiLtJ mice compared with those in controls, which tended to increase with the severity of inflammation in the salivary gland tissues. Moreover, the SDC-1 concentrations in the blood and SMGs of NOD/ShiLtJ mice started to increase significantly at 6 weeks of age (blood, 9.6 ± 1.2 vs. 2.8 ± 0.6, *p* < 0.001; SMG, 4.6 ± 0.6 vs. 1.6 ± 0.4, *p* = 0.002), during which no inflammatory response was detected in the SMGs of NOD/ShiLtJ mice. In the NOD/ShiLtJ mice, SDC-1 concentrations in the blood and SMG of 8-week-old mice tended to increase compared with those of 6-week-old mice, although the differences were not statistically significant. SDC-1 concentrations in the blood and SMGs started to show a significant increase at 10 weeks of age (8-week-old vs. 10-week-old mice: blood, 9.2 ± 1.4 vs. 15.0 ± 1.6, *p* = 0.025; SMG, 6.8 ± 1.3 vs. 11.5 ± 1.3, *p* = 0.047).

### 2.3. Expression of B-Cell Chemokines and Their Receptors in SMG Tissues of NOD/ShiLtJ Mice

We performed a Western blotting assay for these chemokines to identify altered expressions of chemokines in inflamed salivary gland tissues of NOD/ShiLtJ mice (Figure 3A). In 6-week-old NOD/ShiLtJ mice without SMG inflammation, the expressions of IL-7, CCL21, and CXCL13 were significantly upregulated, suggesting that SGECs begin to secrete inflammatory substances such as chemokines before inflammatory cell infiltration. The expression of CXCR4 and CXCR5 also increased in the SMGs of 6-week-old NOD/ShiLtJ mice, but the expression levels were not so high. In 8-week-old mice, a more pronounced increase in the expression of IL-7 and CXCL13 occurred, when salivary gland inflammation began to be observed. The expressions of CXCR4 and CXCR5 were evident from 8 weeks, which is considered to be related to the fact that B cell infiltration increases in earnest in the SMGs of 8-week-old NOD/ShiLtJ mice. Along with the changes in the expression of chemokines, we also identified changes in the expression of SDC-4. SDC-4 was expressed to a similar extent in the SMGs from 6-week-old NOD/ShiLtJ mice as in those of control mice, but was barely detectable in the SMGs from 8-week-old mice, when the inflammatory responses began to develop (Figure 3A). Additionally, infiltration of inflammatory cells, including B cells, T cells, and macrophages, increased with age in the salivary glands of NOD/ShiLtJ mice, along with increased concentrations of chemokines.

### 2.4. Identification of CXCL13 Binding to SDC-1

Although CXCL13 is shown to be functionally presented to its receptor in a HS-bound form [15], whether CXCL13 binds to SDC-1 is not known. To determine whether CXCL13 binds to SDC-1, we first performed co-IF staining with SMG tissues and epithelial cells. The expression of CXCL13 and SDC-1 was upregulated in response to increased inflammation, and CXCL13 and SDC-1 were detected together on the surface of ductal epithelial cells (Figure 4A). Additionally, the co-localization of SDC-1 and CXCL13 was confirmed through co-IF staining using normal murine mammary gland (NMuMG) cells, which are epithelial cell lines derived from normal glandular tissue of mice and which express SDC-1 abundantly on their cell surface (Figure 4B). Subsequently, we investigated whether CXCL13 forms a complex with SDC-1 through an immunoprecipitation (IP) assay using NMuMG cells (Figure 4C). SDC-1 formed a complex with CXCL13, suggesting that CXCL13 binds to SDC-1 directly through the HS on the surface of ductal epithelial cells and participates in the inflammatory pathway through B-cell chemotaxis (Figure 4C).

### 2.5. Improvement of Salivary Gland Inflammation in an HS-Dependent Manner

We examined whether treatment with HS could attenuate inflammation in SMG tissues. NOD/ShiLtJ mice aged 6–10 weeks were treated with HS at 5 mg/kg intraperitoneally three times per week. The degree of tissue inflammation, intensity of inflammatory cell infiltration, and GC formation were assessed in inflamed SMG tissues at the end of 10 weeks of age. The mice that received HS exhibited significant improvement in the inflammation of SMG tissues, as assessed by ratio index, compared with controls (*p* = 0.002, Figure 5A). The formation of GCs decreased (*p* = 0.017, Figure 5B) after HS treatment, and the intensity of B- (*p* = 0.007, Figure 5C) and T-cell (*p* < 0.001, Figure 5D) infiltrations significantly reduced in the HS-treated group compared with controls.

As single cell sequencing or the immunophenotyping of lymphocytes using multicolor flow cytometry could not be performed on salivary gland tissue due to tissue conditions, we sought to identify the subtypes of B and T cells infiltrating the salivary gland tissue using semi-quantitative RT-PCR for cell surface markers or intracellular markers (Appendix A). The B-cell fractions that play an important role in SjD are known as marginal zone (MZ) B cells, memory B cells, and plasma cells [16]. During B-cell differentiation, certain cell surface molecules are observed at different stages of differentiation, making it difficult to define a specific cell surface marker as a specific subtype of B cell using an RT-PCR assay [17]. There are also cell surface markers that are expressed on cells other than B cells, such as CD1d, which is easily observed in MZ B cells but can also be observed on other antigen presenting cells. SDC-1 is predominantly expressed in plasma cells and epithelial cells in adult cells. To identify B-cell subtypes, we used CD23 and CD1d as MZ B-cell markers, CD27 as memory B-cell and plasma-cell markers [17,18,19], and SDC-1 and IRF4 [20] as plasma-cell markers. After HS treatment, CD27 and IRF4 were significantly reduced and SDC-1 also tended to decrease, suggesting a decrease in the infiltration of plasma cells and memory B cells. Both CD23 and CD1d were significantly reduced, indicating that the MZ B-cell fraction was also reduced after treatment. In conclusion, we found that all three B-cell subtypes, which are important in the pathogenesis of SjD, were reduced.

Next, we used cell-specific intracellular markers for subtypes of T cells, including Tbet in Th1 cells, GATA3 in Th2 cells, RAR-related orphan receptor alpha (RORα) in Th17 cells, and B-cell lymphoma-6 (Bcl-6) in follicular T helper (Tfh) cells, which were used to identify changes in infiltrating T-cell subtypes after HS treatment. In this study, Th1 and Th2 subtypes were significantly decreased after HS treatment, but Th17 cells were increased. In addition, Tfh cells tended to decrease, but there was no statistical significance.

To verify and extend this result, we assessed the expression of chemokines and SDCs in SMG tissues (Figure 5E). Whereas the expressions of IL-7, CCL21, and CXCL13 did not differ before and after treatment, the expression of CXCR5 was significantly reduced after HS treatment (*p* = 0.020). The significant decrease in CXCR5 in inflamed tissues was considered to be associated with reduced infiltration of CXCR5-expressing B cells and decreased GC formation. CXCL12 and CXCR4 showed a trend toward decreased levels, but the differences were not statistically significant.

To further analyze the expressions of CXCL13 and CXCR5 in the SMG tissues, we measured these molecules at messenger RNA (mRNA) levels (Appendix A). The transcript expressions of CXCL13 and CXCR5 were significantly reduced in the HS treatment group (CXCL13, *p* = 0.007; CXCR5, *p* = 0.032). The difference between the transcript and the protein expression of CXCL13 is likely related to the fact that the production of CXCL13 decreased after HS treatment, but some of the produced CXCL13 bound to the injected HS and remained in the salivary gland tissue.

After HS treatment, SDC-4 expression was significantly increased, which is consistent with the finding that SDC-4 expression was significantly reduced in inflamed salivary gland tissues compared to normal tissues. SDC-1 expression also tended to decrease after HS treatment, but this was not statistically significant.

## 3. Discussion

In this study, we found that SDC-1 expression was upregulated in the inflamed SMG tissues and blood of NOD/ShiLtJ mice. SDC-1 on the surface of SGECs was involved in the mechanism of B-cell infiltration into salivary gland tissues by binding CXCL13 through the HS chain. Additionally, treatment with HS attenuated salivary gland inflammation with reduced B-cell infiltration, GC formation, and CXCR5 expression.

Cell surface HS influences a multitude of molecules, cell types, and processes relevant to inflammation through binding to chemokines, cytokines, growth factors, morphogens, enzymes, extracellular matrix proteins and glycoproteins, and cell–cell adhesion receptors [9,10,11]. SDCs constitute a major family of cellular HS [9,10,11], and although some SDCs also harbor chondroitin sulfate (CS) chains, the majority of SDCs’ ligand-binding activities have been attributed to its HS moiety [10]. SDCs are expressed in a developmental and cell-type-specific pattern [9,10,11]. SDC-1 is expressed primarily on the epithelial and plasma cells in adult tissues and functions primarily as a co-receptor that catalyzes the interaction between ligands and their respective signaling receptors.

Few studies have reported on the expression of SDC-1 on ductal epithelial cells in normal salivary gland tissue, with one study reporting on HSPG expression in monkey SMG [21]. In this study, HSPGs were expressed in the following order: SDC-4, SDC-1, SDC-2, and glypican. Previously, we showed that SDC-1 was less expressed on the surface of SGECs in normal salivary gland tissue, whereas its expression increased in inflamed minor salivary glands of patients with SjD [14]. In the present study, NOD/ShiLtJ mice, a mouse model of SjD, also had increased SDC-1 expression in salivary gland tissue compared with control mice, suggesting that the inflammatory milieu regulates SDC-1 expression in the pathogenesis of SjD. Activated SGECs in SjD show the dysregulation of the innate immune signaling pathway, such as an increased activation of nuclear factor-ƙB (NF-ƙB) [22]. NF-ƙB has been reported to participate in the upregulation of SDC-1 expression at the transcriptional level in tumorous conditions [23], suggesting that the dysregulation of the innate immune signaling pathway in SGECs drives SDC-1 expression in salivary gland tissues. Additionally, the expression of all transforming growth factor beta (TGF-β) isoforms is upregulated in ducts within the salivary glands of patients with SjD compared with controls [24], and all three TGF-β isoforms are general inducers of epithelial SDC-1 expression [25].

SDC-1 shedding is a highly regulated process that is mediated by sheddases, including matrix metalloproteinase-2 (MMP-2, gelatinase A) [26], MMP-3 [27], MMP-9 (gelatinase B) [28], MMP-14 (MT1-MMP) [29], and a disintegrin and metalloproteinase-17 (AD-AM17) [30], and the expression of MMPs is regulated by cytokines, growth factors, chemokines, and interactions with extracellular matrix proteins [31]. Overexpressed cytokines, chemokines, and growth factors are observed in the inflamed salivary glands of SjD [4] and can upregulate the expression and catalytic activity of SDC-1 sheddases, such as MMP-3 [32,33], MMP-9 [34,35], and ADAM17 [36]. The most studied MMP in the pathogenesis of SjD is MMP-9 [34,35], whose expression is increased by tumor necrosis factor alpha (TNF-α) [37,38], and this increased MMP-9 promotes SDC-1 shedding in the salivary glands of SjD. Additionally, ADAM17, formerly called TNF-α-converting enzyme (TACE), processes precursor TNF-α to release soluble TNF-α and is involved in the pathogenesis of SjD, especially angiogenesis [39,40]. Anti-Ro antibodies exert their pathogenic effects by triggering the ADAM17 (TACE)/TNF-α/NF-κB axis [41], where an increased expression of ADAM17 leads to the release of both SDC-1 ectodomain and TNF-α, which then causes SDC-1 shedding. In series with our pervious study, demonstrating that the SDC-1 concentrations of saliva and blood were higher in patients with SjD than in healthy participants, blood SDC-1 concentrations were elevated in NOD/ShiLtJ mice compared with control mice. These results suggest that the increased shedding of SDC-1 from inflamed salivary glands is a cause of elevated SDC-1 concentrations in the blood.

Epithelial cells are considered a main culprit in the pathogenesis of SjD. SGECs from SjD regulate the infiltration of lymphocytes into the salivary glands and maintain the niche in which lymphocytes are retained and become aggregated by secreting various chemokines, such as CCL21, CCL22, CXCL9, CXCL10, CXCL12, CXCL13, and IL-7, which facilitate infiltrations of T and B cells into salivary glands from blood and the early attraction of various inflammatory immune cell populations [4,42]. Among these chemokines, CXCL9 and CXCL10 are involved in the infiltration of activated T cells by binding to CXCR3 [43], whereas CXCL12 and CXCL13 contribute to the prominent accumulation of CXCR4^+^ CXCR5^+^ memory B cells in the inflamed glands of patients with SjD [44]. CXCL13, CCL21, and CXCL12 belong to the family of homeostatic chemokines [45] and the ectopic expression of CXCL13 and CCL21 has been associated with the progressive acquisition of lymphoid features by the inflammatory foci [46], although CXCL12 seems insufficient to induce the formation of mature lymphoid tissue [47]. IL-7 is a key cytokine involved in T-cell homeostasis and, in the mouse, B-cell homeostasis [48,49]; it exhibits strong chemoattraction for T cells and may be involved in the organization of ectopic lymphoid structures within inflamed salivary glands in SjD [42].

All chemokines bind to the HS chain through positively charged domains, and HS chain binding and the capability to oligomerize are crucial for the activity of several chemokines in vivo [11,50]. The specific chemokines that bind to SDC-1 have not been completely identified; eotaxin, RANTES, and KC bind to SDC-1 [10], whereas CXCL12 binds to SDC-4, but does not form complexes with SDC-1 and SDC-2 [51]. We demonstrated that SDC-1 binds to CXCL13, based on the following considerations. First, SDC-1 and CXCL13 were co-expressed on the surface of SGECs in the SMGs of NOD/ShiLtJ mice. Second, SDC-1 and CXCL13 were detected together on the surface of NMuMG cells. Third, an IP study using NMuMG cells demonstrated that SDC-1 directly binds to CXCL13. In this study, we found that HS treatment improved tissue inflammation and was associated with improved B-cell infiltration and reduced GC formation, suggesting that treatment with HS inhibited B-cell infiltration by interfering with the binding of CXCL13 to SDC-1. T-cell infiltration also significantly decreased after HS treatment, which may be due to HS treatment preventing SDC-1 from binding to substances that can trigger T-cell chemotaxis and proliferation, such as IL-7 [48]. The protein level of CXCL13 did not decrease after HS treatment, while the transcript level was significantly reduced, suggesting that HS treatment reduced CXCL13 production but some of the CXCL13 remained in the salivary gland tissue after binding to the injected HS. Furthermore, we observed that SDC-4 expression was reduced in inflamed salivary gland tissue and then recovered after HS treatment. SDC-4 is expressed ubiquitously, although it is expressed at lower levels than other co-expressed SDCs on any given cell type [10] and acts as an anti-inflammatory [52] or pro-inflammatory [53] substances. Although the role of SDC-4 in the pathogenesis of SjD has not been reported, these results suggest that SDC-4 may act as an anti-inflammatory agent in the salivary gland inflammation of SjD.

This study has a few limitations. First, although SDCs are the primary physiological form of HS on the cell surface, other cell surface HSPGs such as other SDCs and glycosylphosphatidylinositol-anchored glypican may play a role in the mechanism of salivary gland inflammation in SjD. We identified the expression of SDC-4 along with SDC-1, but not other HSPGs. Glypicans are another typical family of cell surface HSPGs that are expressed predominantly in the central nervous system, although except for glypican-2, glypicans are also expressed in non-neural cells [8]. They differ from SDCs in the structure of their core proteins, site of HS attachment, and ligands they bind to. These factors lead to differences in their function as HSPGs [8,54]. A previous study on SMGs in monkeys has shown that the expression levels of glypican were very low [21]. Additionally, no studies have been reported on the role of glypican in SjD or the direct binding of glypican to CXCL13. Therefore, the prediction of the extent to which glypican plays a role in salivary gland inflammation in SjD is difficult, and further identification and comparative analysis of the expression of other HSPGs in salivary gland tissues in mouse models of SjD will help to clarify the role of SDC-1 in the pathogenesis of SD. Second, although the role of the HS chain of SDC-1 in infectious, inflammatory, and neoplastic diseases have been the most studied, studies have also reported the role of the core protein [55,56,57] and the CS chain [58] in these diseases. However, this study did not investigate the binding of the core protein or CS chain to chemokines or their roles in the pathogenesis of SjD. Further studies of the core protein and CS chain are warranted. Third, we performed an IP assay and co-IF staining using NMuMG cells, murine mammary cells that express abundant cell surface SDC-1, rather than primary SGECs derived from NOD/ShiLtJ mice. However, we confirmed that CXCL13 and SDC-1 are co-expressed on the cell surface in the inflamed SMGs of NOD/ShiLtJ mice by double IF staining. Furthermore, using an IP assay, we demonstrated that CXCL13 directly binds to SDC-1.

## 4. Materials and Methods

### 4.1. Reagents and Antibodies

The reagents and their resources were as follows: Immobilon^®^-Ny+ Membrane (charged Nylon, 0.45 µm) was obtained from Millipore (Burlington, MA, USA), and HS from bovine kidney was purchased from Sigma-Aldrich (St. Louis, MO, USA). Recombinant mouse CXCL13 was purchased from R&D Systems (Minneapolis, MN, USA), and Pierce™ Classic Magnetic IP/Co-IP Kit was obtained from Thermo Fisher Scientific (Waltham, MA, USA). The primary antibodies against SDC-1 ectodomain (BD Bioscience, San Jose, CA, USA, and Thermo Fisher Scientific), SDC4 (Santacruz, Dallas, TX, USA), IL-7 (MyBiosource, San Diego, CA, USA), CCL21 (R&D Systems), CXCL12 (Abcam, Cambridge, UK), CXCL13 (Novus Biologicals, Centennial, CO, USA), CXCR4 (Abcam), CXCR5 (ABclonal, Woburn, MA, USA), SDC4 (Santa Cruz Biotechnology, Dallas, TX, USA), CD4 (Abcam), B220 (BD Bioscience, San Jose, CA, USA), F4/80 (Bio-Rad Laboratories, Hercules, CA, USA), rat immunoglobulin G (control; Santa Cruz Biotechnology), GAPDH (Abcam), and β-actin (Sigma-Aldrich) were used.

### 4.2. Animal Experiments

Female NOD/ShiLtJ mice and female C57BL10 mice were purchased from the Jackson Laboratory (Bar Harbor, ME, USA). We used C57BL/10 mice, which have the same Igh1-b allele as NOD/ShiLtJ mice [59] and do not show inflammation of salivary gland tissues even after long-term observation [60], as control mice in this study. Animals were maintained under a conventional zone in a temperature-controlled environment and a 12/12 h light–dark cycle at the Institute of Fatima hospital of Korea and fed standard mouse chow and water ad libitum. All experimental protocols were conducted following NIH guidelines and approved by the Animal Care and Use Committee of Daegu Fatima Hospital (approval number: F-23-02). After completing the study, the mice were anesthetized with 0.25 g/kg avertin and sacrificed with CO_2_ with efforts made to minimize suffering. After intraperitoneal anesthesia with 0.25 g/kg avertin, 0.4 mL of blood was collected via cardiac puncture using a 1 mL syringe and transferred to a K2EDTA tube (BD Bioscience, San Jose, CA, USA) with good shaking to prevent blood clotting. Fifteen minutes after blood collection, plasma was separated by centrifugation at 800× *g* for 15 min at 4 °C, and the tube was labeled with the date and sample name, and stored frozen at −80 °C. SMG tissues were collected after euthanasia with CO_2_ gas and were fixed in 10% neutral-buffered formalin or frozen in liquid nitrogen for further analytical studies. The observational study was conducted in 6- to 12-week-old NOD/ShiLtJ mice and control mice, with SMG and blood samples collected from 6-, 8-, 10-, and 12-week-old mice. For the HS treatment experiment, NOD/ShiLtJ mice aged 6–10 weeks were treated with 5 mg/kg HS intraperitoneally thrice per week, and the therapeutic effects on SMG tissue were evaluated at the end of 10 weeks of age. The number of mice used in observational experiments was five in each of the control and NOD/ShiLtJ mice groups at 6, 8, 10, and 12 weeks of age, and four NOD/ShiLtJ mice each in the placebo and treatment groups in the HS treatment experiments.

### 4.3. Cell Culture

SDC-1 from epithelial and mesenchymal cells differs significantly in disaccharide composition and in the size and frequency of iduronate-rich hypersulfated regions of HS chains [11,61]. NMuMG cell is an epithelial cell that was isolated from the normal mammary glands of mice and which expresses SDC-1 abundantly on the surface. NMuMG cells were purchased from ATCC (American Type Culture Collection, Manassas, VA, USA) and cultured in Dulbecco’s modified Eagle medium (DMEM) containing 10% fetal bovine serum (FBS) and antibiotics. The cells were starved for 4 h in DMEM containing 0.5% FBS when they reached 80–90% confluency.

### 4.4. Histology and Immunohistochemistry (IHC) Staining

SMG tissues were harvested, fixed in 10% neutral-buffered formalin for 48 h, and embedded in paraffin. Next, 3 μm thick sections of formalin-fixed and paraffin-embedded SMG tissues were stained with hematoxylin and eosin (H&E). For the quantitation of inflammation, H&E-stained slides were observed under the microscope, and image analysis was performed using ImageJ. The intensity of inflammation was quantified using the ratio index, as previously described [62]. An aggregate of >50 mononuclear cells within the salivary gland parenchyma was identified as a lymphocytic focus. The total area of the salivary gland and the regions occupied by lymphocytic foci were measured. Since none of the sections showed fatty infiltration, no areas within the salivary glands were excluded from the analysis. The results are expressed as the percentage of inflammation area, calculated as (area of inflammation ÷ total glandular area of the section) × 100.

For IHC analysis, antigen retrieval was performed at 60 °C overnight in ethylenediaminetetraacetic acid buffer (pH 9.0) for CD4 or with 10 mM sodium citrate (pH 6.0) for F4/80. For B220, no antigen retrieval was needed. Endogenous peroxidase activity was depleted by treating with 0.3% H_2_O_2_, and non-specific binding was blocked with 5% bovine serum albumin. The sections were incubated with primary antibodies at appropriate dilutions, followed by treatment with biotinylated secondary antibodies and Vectastain Elite ABC reagents (Vector Laboratories, Burlingame, CA, USA). Semiquantitative analyses of B- and T-cell infiltrations were performed after conducting deconvolution and downstream analysis using Fiji (https://imagej.net/Fiji/Downloads, accessed on 15 July 2024).

### 4.5. IF Staining

Co-IF staining of SMG tissue was performed for SDC1 and CXCL13 or B220 and CXCR5. Briefly, the sections were deparaffinized, rehydrated, retrieved with sodium citrate buffer (10 mM sodium citrate, 0.05% Tween 20, pH 6.0), and incubated with each primary antibody (rabbit polyclonal antibodies against SDC1: #36-2900, goat polyclonal antibody for CXCL13: AF470, goat anti-mouse IgG (H + L) against B220: AF488, and goat anti-rabbit IgG (H + L) against CXCR5: AF594). Bound antibodies were visualized using fluorescence-conjugated secondary antibodies. For the semiquantitative analysis of GC, the area of inflammatory areas with DAPI was measured using ImageJ. The area of the B220^+^CXCR5^+^ cells was obtained by measuring the number of cells using Plugins-analyse-cell counter Fiji software (ImageJ). The GC area was then expressed as the percentage of B220^+^CXCR5^+^ area divided by the total inflamed area.

### 4.6. Western Blotting Analysis

SMG tissues were lysed in the T-PER™ Tissue Protein Extraction Reagent (Thermo, 1:20 *w*/*v*) and then centrifuged at 16,000× *g* for 20 min at 4 °C. The supernatant was collected and the protein concentration was determined spectrophotometrically by Bradford’s protein assay (Bio-Rad). Protein (15 μg per lane) and pre-stained standards (BioRad Laboratories, Hercules, CA, USA) were loaded onto a 15% sodium dodecyl sulfate-polyacrylamide gel and separated by electrophoresis. After electrophoresis, the resolved proteins were transferred from gel to a polyvinylidene fluoride membrane [63]. A blot buffer [25 mM Tris/192 mM glycine, pH 8, 20% (*v*/*v*) methanol] was used for gel and membrane saturation and blotting. The blot conditions were the following: 400 mA (constant amperage), 100 V for 90 min. Blots were then blocked by Tris-buffered saline (TBS) buffer [20 mM Tris/150 mM NaCl, pH 7.4] with 0.1% (*v*/*v*) Tween 20, 5% *w*/*v* non-fat dried milk at room temperature for 1 h, and washed three times with 0.1% (*v*/*v*) Tween 20–TBS 1X. Membranes were then incubated overnight at 4 °C with primary antibodies against IL-7, CCL21, CXCL12, CXCL13, CXCR4, CXCR5, SDC4, GAPDH, and β-actin and were incubated with secondary antibodies at room temperature for 1 h. Immunoreactivities were detected by enhanced chemiluminescence (ECL) reagents. The protein-specific signals were measured using Q9-Alliance software (version 18.16c; Uvitec, Cambridge, England, UK).

### 4.7. Dot-Blotting Analysis

Equal amounts of protein (5 μg per dot) and standards diluted in acetic acid buffered saline-Tween 20 solution were placed in a dot blot apparatus and transferred to immobilonNy+ membrane. By acidifying the samples, only highly anionic molecules, such as SDC-1, were retained by the cationic ImmobilonNy+ membrane, while most proteins passed through the membrane during dot blotting [25]. The membranes were incubated with blocking buffer [50 mM Tris/150 mM NaCl, pH 7.5] with 0.1% (*v*/*v*) Tween 20, 10% *w*/*v* non-fat dried milk at room temperature for 1 h and then incubated at 4 °C overnight with rat anti-mouse SDC-1 ectodomain antibodies. The membranes were incubated with secondary antibodies (normal rat immunoglobulin G) at room temperature for 2 h, and immunoreactivities were detected using ECL reagents. Protein-specific signals were measured using Q9-Alliance software.

### 4.8. IP Assay

NMuMG cells were cultured in DMEM containing 10% FBS and antibiotics. The cells were starved in DMEM containing 0.5% FBS for 4 h when they reached 80–90% confluency. Then, cells were treated with 0, 50, 100, and 150 nM recombinant mouse CXCL13 for 18 h. The cell pellets of NMuMG cells were collected following the protocol outlined in the Pierce™ Classic Magnetic IP/Co-IP Kit (Thermo, #88804). To combine the samples, cell lysate (500 μg) and IP anti-SDC-1 antibody (5 µg) were added to an E-tube, diluted with IP lysis/wash buffer, and incubated overnight at 4 °C for the immune complex. The antigen–antibody mixture was added to an E-tube containing pre-washed protein A/G magnetic beads, mixed, and incubated at room temperature for 1 h. Then, the beads with a magnetic stand were collected and the unbound samples were removed and saved for analysis. For alternative elution, lane marker sample buffer was added to the tube, and it was incubated at room temperature with mixing for 10 min to magnetically separate the beads. The mouse anti-SDC-1-antibody binding complexes were analyzed using Western blotting antibodies against CXCL13 or rat immunoglobulin G as the control.

### 4.9. Semi-Quantitative Reverse Transcription PCR (RT-PCR)

Total RNA was isolated from salivary gland tissues using TRIzol reagent (Invitrogen, Carlsbad, CA, USA), and the first-strand cDNA was synthesized using Superscript III reverse transcriptase (Invitrogen). Real-time PCR was performed using a ViiA™ 7 Real-Time PCR System (Applied Biosystems, Carlsbad, CA, USA) using SYBR^®^ Green Master Mix (Applied Biosystems). The oligonucleotides used are listed as follows: CXCL13: F (GGCCACGGTATTCTGGAAGC), R (ACCGACAACAGTTGAAATCACTC), CXCR5: F (GAATGACGA CAGAGGTTCCTG), R (GCCCAGGTTGGCTTCTTAT), CD27: F (AGAAGAAACCACGGGCCAAAT), R (CTCCTGGAT AGGGATAGCACTG), SDC-1: forward (F) (TCAGAGCCTTTTGGACAGGAA), reverse (R) (CACCAGGCACACAGCAAAGA), IRF4: F (AAAGGCAAGTTCCGAGAAGGG), R (CTCGACCAA TTCCTCAAAGTCA), CD23: F (ATCTCAGCCGTGATCTTGTTCT), R (TCGACAGTAGAGTAGGGTA AGGA), CD1d: F (CTGTCTGCGGGCTGTGAAAT), R (TCCCCAGAATCTCACGACATATT), CD3: F (AAGCCTGTGACCCGAGGAA), R (TGCGGATGGGCTCATAGTCT), CD4: F (AAGTGACCTTCAG TCCGGGTA), R (GGGTTAGAGACCTTAGAGTTGCT), Th1 (T-bet): F (AACCGCTTATATGTCCA CCCA), R (CTTGTTGTTGGTGAGCTTTAGC), Th2(GATA-3): F (CGGAAGAGGTGGACGTACTTTT), R (CGTAGCCCTGACGGAGTTTC), Th17(RORα): F (CAGAGCAATGCCACCTACTCCT), R (CTG CTT CTTGGACATCCGACCA), Bcl-6: F (CCGGCACGCTAGTGATGTT), R (GCACTGTCTTATGGGCTC TAAAC) and β-actin: F (CTAAGGCCAACCGTGAAAAG), R (ACCAGAGGCATACAGGGACA). All the RT-qPCR reactions were performed in triplicate. The expression levels of interested genes were normalized to geometric means of β-actin as the internal control, and calculated using the 2^−ΔΔCT^ method.

### 4.10. Statistical Analysis

Data and statistical analyses adhered to recommendations on experimental design and analysis. All quantitative values were expressed as mean ± standard error of the mean. The statistical significance of differences between NOD/ShiLtJ and control mice was analyzed by Student’s *t*-test. All statistical analyses were performed at two-sided α = 0.05 significance level using SPSS Statistics for Windows version 21.0 (IBM, Armonk, NY, USA).

## 5. Conclusions

We found that SDC-1 expression was upregulated in the inflamed SMG tissues and blood of NOD/ShiLtJ mice. SDC-1 on the surface of SGECs was involved in the mechanism of B-cell infiltration into salivary gland tissues by binding CXCL13 through the HS chain. Additionally, treatment with HS attenuated salivary gland inflammation by reducing B-cell infiltration, GC formation, and CXCR5 expression. Decreased CXCR5 expression after HS treatment may be associated with decreased CXCR5-expressing inflammatory cells, including CXCR5^+^ memory B cells. In conclusion, SDC-1 possibly plays an important role in the pathogenesis of SjD by binding CXCL13 via the HS chain to induce B-cell chemotaxis into salivary gland tissues. SDC-1 on the surface of SGECs was involved in the mechanism of B-cell infiltration. Further studies with mouse models of SjD with a knockout background of the SDC-1 gene may provide additional clues as to how SDC-1 plays a role in the inflammatory mechanisms of SjD.

## Figures and Tables

**Figure 1 ijms-25-09375-f001:**
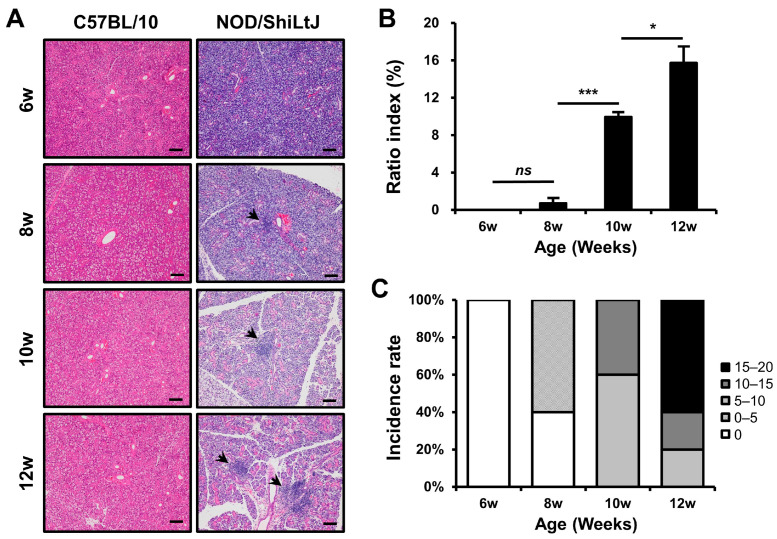
Histopathologic changes in the submandibular glands (SMGs) from NOD/ShiLtJ mice. (**A**,**B**) Hematoxylin and eosin-stained sections of the SMGs show focal inflammatory foci (indicated by black arrow heads). The severity of inflammation of the SMGs is assessed by ratio index ([area of inflammation/total glandular area of the section] × 100). (**C**) Both incidence rate and severity of inflammation in the SMGs show an increase with age. Inflammatory changes can be observed in all salivary gland tissues of 10- and 12-week-old mice and are more intense in 12-week-old mice than in 10-week-old mice. n in each group = 5. Bars show the mean and S.E.M for each group. ns = not significant; * = *p* < 0.05, *** = *p* < 0.001. Scale bar = 100 µm.

**Figure 2 ijms-25-09375-f002:**
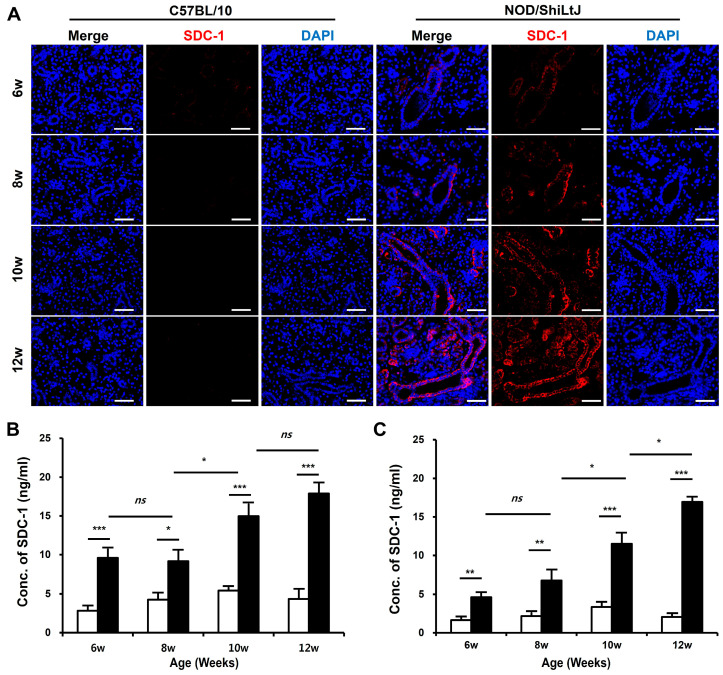
Expression of syndecan-1 (SDC-1) in blood and submandibular gland (SMG) tissues of NOD/ShiLtJ mice. (**A**) Immunofluorescence (IF)-stained sections of the SMGs show that the expression of SDC-1 increases with the severity of inflammation in the ductal epithelium in the SMGs of NOD/ShiLtJ mice. Tissue sections were stained with anti-SDC-1 mAb (red) and counterstained with 4′,6-dia-midino-2-phenylindole (DAPI, blue). Scale bar = 50 µm. (**B**,**C**) SDC-1 concentrations in the (**B**) blood and (**C**) SMG of NOD/ShiLtJ mice are significantly higher than in those of controls. SDC-1 concentrations were measured using dot blotting. n in each group = 5. Bars show the mean and S.E.M for each group. ns = not significant; * = *p* < 0.05, ** = *p* < 0.01, *** = *p* < 0.001. ■ NOD/ShiLtJ mouse, □ C57BL/10 mouse.

**Figure 3 ijms-25-09375-f003:**
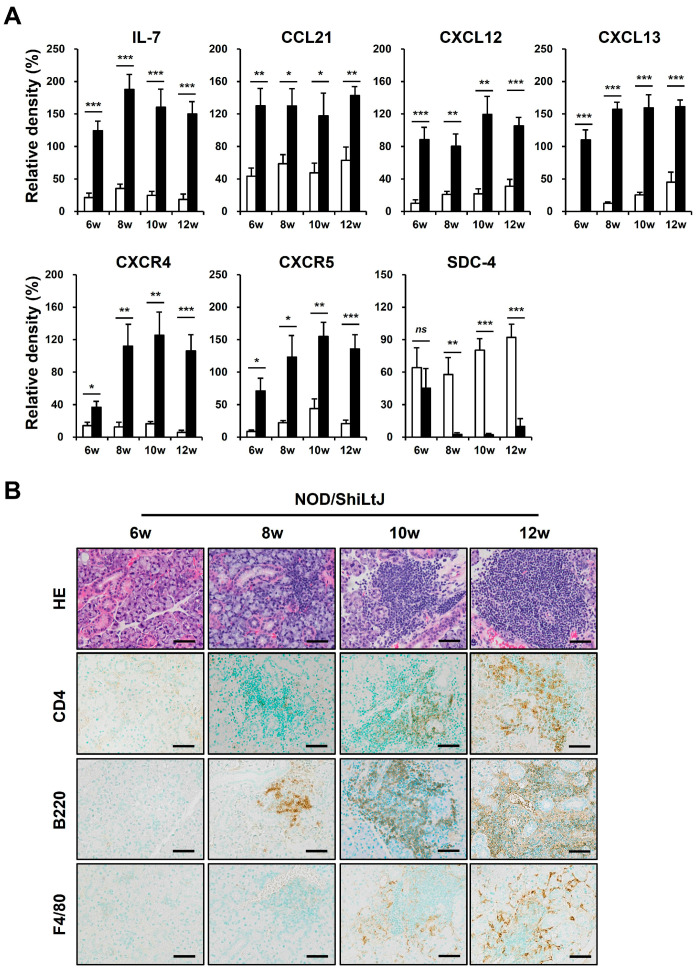
Upregulated chemokine expression and inflammatory cell infiltration in submandibular glands (SMGs) of NOD/ShiLtJ mice. (**A**) Western blotting assay for chemokines and chemokine receptors. n in each group = 5. Bars show the mean and S.E.M for each group. ns = not significant; * = *p* < 0.05, ** = *p* < 0.01, *** = *p* < 0.001. ■ NOD/ShiLtJ mouse, □ C57BL/10 mouse. (**B**) Representative histopathologic findings of SMGs. Upper panel: H&E-stained salivary glands. Lower panels: immunohistochemical staining for CD4, B220, and F4/80 (brown). Scale bar = 50 µm.

**Figure 4 ijms-25-09375-f004:**
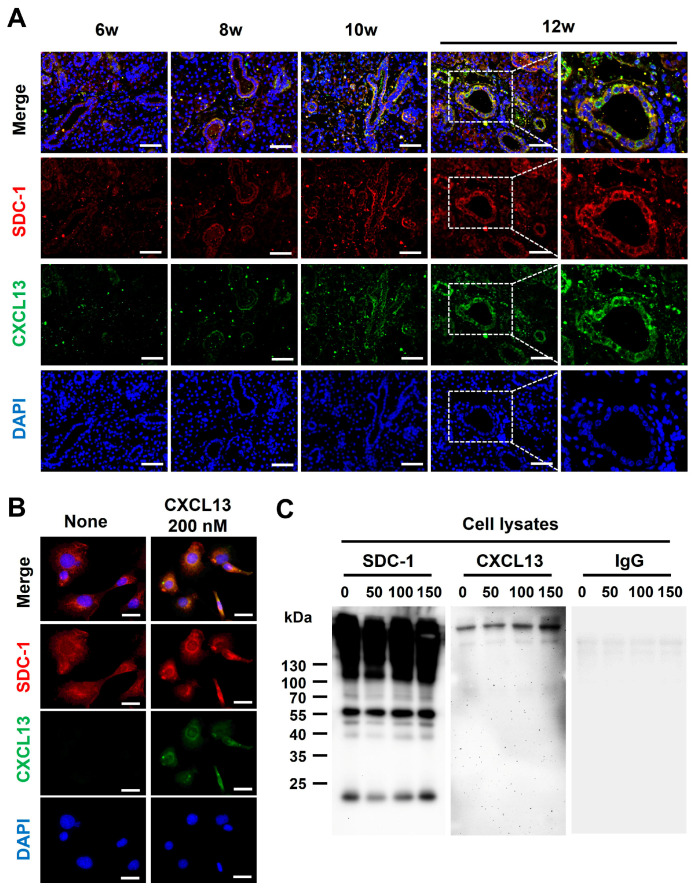
Identification of CXCL13 binding to syndecan-1 (SDC-1). (**A**) Co-expression of SDC-1 and CXCL13 in submandibular gland tissues of NOD/ShiLtJ mice. Each section was double-stained with immunofluorescent antibodies to determine the distribution of SDC-1 (CD138, red) and CXCL13 (green) on the epithelial cells. The sections were counterstained with 4′,6-diamidino-2-phenylindole (DAPI, blue). The merged images obtained by confocal microscopy show the yellow co-localization of both molecules. Magnified views of the boxed areas are shown in the right column. Scale bars = 50 and 25 µm. (**B**) Co-expression of SDC-1 and CXCL13 on normal murine mammary gland (NMuMG) cells. NMuMG cells were double-stained with anti-CXCL13 mAb (green) and anti-SDC-1 mAb (red). The sections were counterstained with DAPI (blue). Scale bar = 25 µm. (**C**) Association of SDC-1 with CXCL13 at the cell surface of NMuMG cells. NMuMG cells were incubated with 0, 50, 100, and 150 nM recombinant mouse CXCL13 for 18 h. The cell pellets of NMuMG cells were harvested following the protocol outlined in the Pierce™ Classic Magnetic IP/Co-IP Kit. The cell lysates were incubated with mouse anti-SDC-1 antibody, and anti-SDC-1 antibody binding complexes were analyzed by Western blotting assay with antibody against CXCL13 (Novus Biologicals, Centennial, CO, USA) or rat immunoglobulin G (control, Santacruz, Dallas, TX, USA).

**Figure 5 ijms-25-09375-f005:**
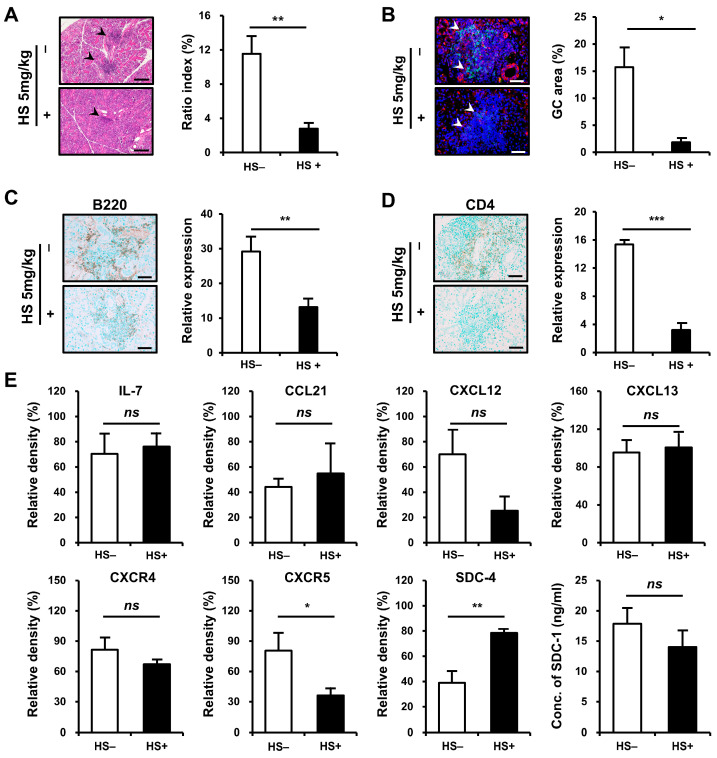
Therapeutic efficacy of heparan sulfate (HS) treatment on inflammatory cell infiltration in the submandibular gland (SMG) of NOD/ShiLtJ mice. Female NOD/ShiLtJ mice were treated with either a phosphate-buffer solution (n = 5) or HS (5 mg/kg, 3 times/week) (n = 5) for 5 weeks beginning at 6 weeks of age. (**A**) Representative histopathologic findings (**left panels**) and histologic scores (ratio index, **right**) of SMG sections from controls and HS-treated mice. Focal inflammatory foci are indicated by black arrow heads. Scale bar = 100 μm. (**B**) Representative immunofluorescence staining for germinal center (GC, **left panels**) and measurement of GC area (**right**) tissue sections which were double-stained with anti-CXCR5 mAb (red) and anti-B220 mAb (green). The sections were counterstained with 4′,6-diamidino-2-phenylindole (DAPI, blue). Semiquantitative determination of T-cell infiltration was performed using Fiji (https://imagej.net/Fiji/Downloads, accessed on 15 July 2024). GCs are indicated by white arrow heads. Scale bar = 50 μm. (**C**) Representative immunohistochemical (IHC) staining for B cells (**left panels**) and measurement of B-cell infiltrates (**right**). Positivity is indicated by diaminobenzidine (DAB) precipitation (brown). Semiquantitative determination of B-cell infiltration was performed using Fiji (ImageJ). Scale bar, 50 μm. (**D**) Representative IHC staining for helper T cells (**left panels**) and measurement of T-cell infiltrates (**right**). Positivity indicated by DAB precipitation (brown). Semiquantitative determination of T-cell infiltration was performed using Fiji (ImageJ). Scale bar = 50 μm. (**E**) Western blotting assay for chemokines, chemokine receptors, and syndecans 4 and SDC-1. n in each group = 4. Bars show the mean and S.E.M for each group. ns = not significant; * = *p* < 0.05, ** = *p* < 0.01, *** = *p* < 0.001. □ placebo-treated group, ■ HS-treated group.

## Data Availability

The data presented in this study are available on request from the corresponding author.

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
