# Peer review of "Syndecan-1 Plays a Role in the Pathogenesis of Sjögren’s Disease by Inducing B-Cell Chemotaxis through CXCL13–Heparan Sulfate Interaction"

_ijms, 2024, doi:10.3390/ijms25179375_

Round 1

Reviewer 1 Report

Comments and Suggestions for Authors

Sjogren’s patients exhibit lymphocytic infiltration within salivary, lachrymal and other secretory glands which interferes with glandular function and cause the characteristic sicca symptoms.  B-cells play a central role in the pathogenesis, becoming activated and functioning as cytokine secreting, antigen presenting and autoantibody producing cells. Many of the proposed treatments in development target B-cell function.

Epithelial cells within the salivary gland secrete chemokines which induce B cell chemotaxis. Syndecan-1 (SDC-1) is a heparan sulphate proteoglycan (HSPG) expressed on epithelial cells. On the cell surface SDC-1 binds to and regulates Heparan Sulphate (HS) binding molecules which include chemokines, cytokines, growth factors, and extra-cellular matrix components. SDC-1 can modulate inflammation acting as both pro and anti-inflammatory. The authors have previously demonstrated increased SDC-1 expression within the salivary glands of patients with Sjogren’s and have found raised concentrations of SDC-1 in saliva and blood of patients with Sjogren’s compared to healthy controls.

In this study the authors have examined the effects of SD-1 in early Sjogren’s and its role in inflammation induction within the salivary glands in Sjogren’s.

They used a mouse model to explore their hypothesis. They noted increased SD-1 concentrations within the submandibular salivary gland and blood preceding inflammation within the salivary glands.

They confirmed that in this mouse model inflammation within the salivary glands was not visible at 6 weeks, started to appear from 8 weeks and was universal by 10 weeks.  They identified SDC-1 expression within the glands which increased as tissue inflammation worsened and was predominantly expressed on salivary duct epithelial cells. They also found that circulating SDC-1 in the blood stream increased with increasing inflammation. There was a significant increase in blood concentration of SDC-1 from 6 weeks, preceding visible inflammation within the salivary glands.

Western blot assays showed upregulation of inflammatory chemokines IL-7, CCL21 and CXCL13 in salivary gland tissues at 6 weeks, preceding inflammatory cell infiltration. This expression was more pronounced by 8 weeks.

The authors found co-localization of SDC-1 and CXCL13 on ductal epithelial cells and was also found to form a complex on immunoprecipitation assay suggesting direct binding of the two.

Finally, the authors looked at whether treatment with HS could reduce the inflammation within the salivary gland tissues. They report a reduction in the severity of inflammation and a reduction in CXCR5 expression in the treated mice.

Issues

Many authors have moved from referring to Sjogren’s as Sjogren’s syndrome to calling it Sjogren’s disease (see Sjögren's Disease, Not Syndrome - PubMed (nih.gov)). The authors may wish to consider this.

Can the authors reference their statement referring to their previous work on SDC-1 (page 2 , line 67).

The ‘Materials and Methods’ section should include reference to the number of mice examined, numbers treated with HS etc.

Author Response

  1. Many authors have moved from referring to Sjogren’s as Sjogren’s syndrome to calling it Sjogren’s disease (see Sjögren's Disease, Not Syndrome - PubMed (nih.gov)). The authors may wish to consider this.

[Response] We appreciate the reviewer’s suggestion.

Many recent papers have used the term Sjögren's disease (SD) instead of Sjögren's syndrome (SS). We are aware of the difference in meaning between syndrome and disease and have modified primary SS to SD in this paper.

  1. Can the authors reference their statement referring to their previous work on SDC-1 (page 2 , line 67).

[Response] We thank the reviewer for the constructive comment.

In the “Discussion” of the submitted manuscript, we had described our results on the expression of syndecan-1 (SDC-1) in the salivary glands of mouse model of SD with reference to previous our data on the expression of SDC-1 in the salivary glands of patients with SD as follows: Previously, we showed that SDC-1 was lowly expressed on the surface of SGECs in normal salivary gland tissue, whereas its expression increased in inflamed minor salivary glands of patients with pSS [17]. In the present study, NOD/ShiLtJ mice, a mouse model of pSS, also had increased SDC-1 expression in salivary gland tissue compared with control mice, suggesting that the inflammatory milieu regulates SDC-1 expression in the pathogenesis of pSS.

We referenced our previous data on the expression of SDC-1 in the blood and updated our manuscript as follows: “In series with our pervious study demonstrating the SDC-1 concentrations of saliva and blood were higher in patients with SD than in healthy participants, blood SDC-1 concentrations were elevated in NOD/ShiLtJ mice compared with in control mice. These results suggest that increased shedding of SDC-1 from inflamed salivary glands is a cause of elevated SDC-1 concentrations in the blood.”

  1. The ‘Materials and Methods’ section should include reference to the number of mice examined, numbers treated with HS etc.

[Response] We thank the reviewer for the kind feedback.

We had previously described the number of mice used in each experiment in the legends of the figures. We added a more detailed explanation in the “Materials and Methods” as follow: “the number of mice used in observational experiments was five in each of the control and NOD/ShiLtJ mice groups at 6, 8, 10, and 12 weeks of age, and four NOD/ShiLtJ mice each in the placebo and treatment groups in the HS treatment experiments.

Reviewer 2 Report

Comments and Suggestions for Authors

ž   Line 22: I think ‘autoimmune epithelitis’ as another name of SS is not necessary because it is not widely accepted, and the term does not reflect well the characteristics of SS which might affect organs of whole body.

ž   I assume the content of line 67 to 69 corresponds to reference 17. If correct, the authors can cite reference 17 there. 

ž   Figure 4C: What does the red arrow mean and where can I find positive results of CXCL13?

ž   How do the authors think of the dissimilar results of CXCL13 between Figure 4 and 5?

ž   Materials and Methods section: How did the authors collect blood samples and store properly?

Author Response

  1. Line 22: I think ‘autoimmune epithelitis’ as another name of SS is not necessary because it is not widely accepted, and the term does not reflect well the characteristics of SS which might affect organs of whole body.

[Response] We thank the reviewer for the comment.

The authors have removed the words 'autoimmune epithelitis' as per the reviewer's comment.

  1.  I assume the content of line 67 to 69 corresponds to reference 17. If correct, the authors can cite reference 17 there. 

[Response] We thank the reviewer for the kind feedback.

The content of this paper in lines 67 to 69 is from our previous paper on the SDC-1 levels in patients with Sjogren’s disease and we have cited this paper as per the reviewer’s comment.

  1. Figure 4C: What does the red arrow mean and where can I find positive results of CXCL13?

[Response] We appreciate the reviewer’s critical comment. In this study, we investigated whether CXCL13 forms a complex with SDC-1 by immunoprecipitation (IP) assay. For IP experiment, cell lysates were incubated with anti-syndecan-1 (SDC-1) antibody and then anti-SDC-1 complexes collected on beads were characterized by western blotting assay with anti-SDC-1 or anti-CXCL13 antibodies or rat IgG for negative control. As shown in Figure 4C, the complexes were immunoreactive to anti-SDC-1 or anti-CXCL13 antibodies, but not to IgG isotype. Based on the fact that SDC-1 binds to chemokines through heparan sulfate (HS), these results suggest that SDC-1 binds directly to CXCL13 through HS. In the previously submitted Figure 4C, we had left out the negative control IgG lane to fit the figure arrangement. We have modified the figure to include the IgG lane to show the clear results of the IP assay.

In previous study [1], SDC-1 was identified at around 56 kD by western blotting assay after treatment with heparitinase and chondroitinase ABC to remove glycosaminoglycan (GAG) chains, but without enzyme treatment, the molecular weight was observed around 200 kD due to GAG chains. In our study, all immunoreactive proteins identified at least 100 kD and up to a larger molecular size were SDC-1 because we did not treat these enzymes. In the previous Figure 4C, we had put a red arrow to indicate the minimum molecular weight of SDC-1 proteins with GAG chains. In the revised figure, the arrow has been removed.

Reference

  1. Kim, C.W.; Goldberger, O.A.; Gallo, R.L.; Bernfield, M. Members of the syndecan family of heparan sulfate proteoglycans are expressed in distinct cell-, tissue-, and development-specific patterns. Mol.Biol.Cell. 1994, 5, 797-805. https://doi: 10.1091/mbc.5.7.797

  1. How do the authors think of the dissimilar results of CXCL13 between Figure 4 and 5?

[Response] We fully acknowledge the reviewer's comment that the CXCL13 results in Figures 4 and 5 are different.

We had predicted that CXCL13 would be reduced in the HS treatment group after HS treatment. However, we had found that CXCR5 was significantly reduced, whereas CXCL13 was not significantly changed between the treatment and control groups. We deduced the reason for this result and explained it in the “Discussion” as follows: Although salivary gland inflammation improved after HS treatment, the concentrations of chemokines, such as CXCL13 and IL-7, did not decrease significantly in inflamed salivary gland tissues, which may be due to the HS chain staying in the salivary gland tissue after binding to the chemokines. To prove our hypothesis, we measured the mRNA levels of CXCL13, CXCR5, and SDC-1 in SMG tissue, as mRNA levels are an indicator of protein synthesis. We found that the transcript expression of CXCL13 and CXCR5 was significantly reduced in HS treatment group (CXCL13, P = 0.00681; CXCR5, P = 0.031985) and that of SDC-1 showed a tread toward decreased level (P = 0.265).

We updated our manuscript with additional data to “Results” and adding a supplementary figure as follows: “to further analyze the expression of CXCL13, and CXCR5 in the SMG tissues, we measured these molecules at messenger RNA (mRNA) levels (Figure S1). The transcript expression of CXCL13 and CXCR5 was significantly reduced in HS treatment group (CXCL13, P = 0.007; CXCR5, P = 0.032). The difference between the transcript and protein expression of CXCL13 is likely related to the fact that the production of CXCL13 decreased after HS treatment, but some of produced CXCL13 bound to the injected HS and remained in the salivary gland tissue.

Figure S1. Transcript level of CXCL13 and CXCR5 in the submandibular glands of control and heparin sulfate-treated NOD.ShiLtJ mice.

Additionally, we have modified what we described in the “Discussion” as follow: “The protein level of CXCL13 did not decrease after HS treatment, while the transcript level was significantly reduced, suggesting that HS treatment reduced CXCL13 production but some of the CXCL13 remained in salivary gland tissue after binding to the injected HS.”

The semi-quantitative reverse transcription PCR experiments have been described in the “Materials and Methods” as follows: “total RNA was isolated from salivary gland tissues using TRIzol regent (Invitrogen, Carlsbad, CA), and the first strand cDNA was synthesized using Superscript III reverse transcriptase (Invitrogen). Real-time PCR was performed using a ViiA™ 7 Real-Time PCR System (Applied Biosystems, CA) using SYBR® Green Master Mix (Applied Biosystems). The oligonucleotides used are listed as follows: CXCL13: F (GGCCACGGTATTCTGGAAGC), R (ACCGACAACAGTTGAAATCACTC), CXCR5: F (GAATGACGA CAGAGGTTCCTG), R (GCCCAGGTTGGCTTCTTAT), CD27: F (AGAAGAAACCACGGGCCAAAT), R (CTCCTGGAT AGGGATAGCACTG), SDC-1: forward (F) (TCAGAGCCTTTTGGACAGGAA), reverse (R) (CACCAGGCACACAGCAAAGA), IRF4: F (AAAGGCAAGTTCCGAGAAGGG), R (CTCGACCAA TTCCTCAAAGTCA), CD23: F (ATCTCAGCCGTGATCTTGTTCT), R (TCGACAGTAGAGTAGGGTA AGGA), CD1d: F (CTGTCTGCGGGCTGTGAAAT), R (TCCCCAGAATCTCACGACATATT), CD3: F (AAGCCTGTGACCCGAGGAA), R (TGCGGATGGGCTCATAGTCT), CD4: F (AAGTGACCTTCAG TCCGGGTA), R (GGGTTAGAGACCTTAGAGTTGCT), Th1 (T-bet): F (AACCGCTTATATGTCCA CCCA), R (CTTGTTGTTGGTGAGCTTTAGC), Th2(GATA-3): F (CGGAAGAGGTGGACGTACTTTT), R (CGTAGCCCTGACGGAGTTTC), Th17(RORα) : F (CAGAGCAATGCCACCTACTCCT), R (CTG CTT CTTGGACATCCGACCA), Bcl-6: F (CCGGCACGCTAGTGATGTT), R (GCACTGTCTTATGGGCTC TAAAC) and β-actin: F(CTAAGGCCAACCGTGAAAAG), R (ACCAGAGGCATACAGGGACA). All the RT-qPCR reactions were performed in triplicate. The expression levels of interested genes were normalized to geometric mean of b-actin as internal control, and calculated by 2ΔDDCT method.

  1. Materials and Methods section: How did the authors collect blood samples and store properly?

[Response] We thank the reviewer for the comment.

We have described blood sampling and storage in the “Materials and Methods” as follows: “After intraperitoneal anesthesia with 0.25 g/kg avertin, 0.4 ml was collected via cardiac puncture using a 1 ml syringe and transferred to a K2EDTA tube (BD Bioscience, San Jose, CA) with good shaking to prevent blood clotting. Fifteen minutes after blood collection, serum was separated by centrifugation at 3000 rpm for 15 minutes at 4°C, and the tube was labeled with the date and sample name, and stored frozen at -80°C.”

Reviewer 3 Report

Comments and Suggestions for Authors

A potential interesting paper, well-written. The exact conclusions should be stated in the end of the abstract. Moreover, please provide exact p-values in the whole manuscript, especially on Figures. Limitations of the study may results in a bias with a huge impact on final implications. 

Comments on the Quality of English Language

Minor editing of English language required.

Author Response

A potential interesting paper, well-written. The exact conclusions should be stated in the end of the abstract. Moreover, please provide exact p-values in the whole manuscript, especially on Figures. Limitations of the study may results in a bias with a huge impact on final implications. 

[Response] We appreciate the reviewer’s critical comments.

We have updated some parts of abstract and conclusion as follows: “Furthermore, NOD/ShiLtJ mice treated with 5 mg/kg HS intraperitoneally thrice a week during 6–10 weeks of age attenuated salivary gland inflammation with reduction of B-cell infiltration, GC formation and CXCR5 expression. These findings suggest that SDC-1 plays pivotal roles in the pathogenesis of SD by binding to CXCL13 through the HS chain.”

We have updated our manuscript in the “Conclusions” as follows: “We found that SDC-1 expression was upregulated in inflamed SMG tissues and blood of NOD/ShiLtJ mice. SDC-1 on the surface of SGECs was involved in the mechanism of B-cell infiltration into salivary gland tissues by binding CXCL13 through HS chain. Additionally, treatment with HS attenuated salivary gland inflammation by reducing B-cell infiltration, GC formation, and CXCR5 expression. Decreased CXCR5 expression after HS treatment may be associated with decreased CXCR5-expressing inflammatory cells, including CXCR5+ memory B cells. In conclusion, SDC-1 possibly plays an important role in the pathogenesis of SD by binding CXCL13 via the HS chain to induce B-cell chemotaxis into salivary gland tissues. SDC-1 on the surface of SGECs was involved in the mechanism of B-cell infiltration. Further studies with mouse models of SD with a knockout background of the SDC-1 gene may be provide additional clue as to how SDC-1 plays a role in the inflammatory mechanism of SD.”

Additionally, we tried to write down the P values in the figures, but Figures 1 and 2 are very graphically complex and it was not possible to write down all the P values. Therefore, we summarize all protein concentrations and P values from this study in the supplementary tables. It would be grateful if reviewer could understand this condition.

Reviewer 4 Report

Comments and Suggestions for Authors

I read this manuscript by Lee et al. with interest, where the authors found that SDC-1 expression was upregulated in inflamed SMG tissues and blood of NOD/ShiLtJ mice. SDC-1 on the surface of SGECs was involved in the mechanism of B-cell infiltration into salivary gland tissues by binding CXCL13 through HS chain. Additionally, treatment with HS attenuated salivary gland inflammation with reduced B-cell infiltration, GC formation, and CXCR5 expression. Here are my comments.

1. Figure 1A and Figure 2A had no wild type control. Please provide.

2. In Figure 2B, the authors measured the changes in SDC-1 concentrations in the blood, how about the IgG production levels in the blood? The relationship between IgG levels and SDC-1 levels?

3. Did the authors assess the B cell and T cell subset changes in SMG of NOD/ShiLtJ mice before and after the HS treatment?

Author Response

I read this manuscript by Lee et al. with interest, where the authors found that SDC-1 expression was upregulated in inflamed SMG tissues and blood of NOD/ShiLtJ mice. SDC-1 on the surface of SGECs was involved in the mechanism of B-cell infiltration into salivary gland tissues by binding CXCL13 through HS chain. Additionally, treatment with HS attenuated salivary gland inflammation with reduced B-cell infiltration, GC formation, and CXCR5 expression. Here are my comments.

  1. Figure 1A and Figure 2A had no wild type control. Please provide.

[Response] We appreciate the reviewer’s critical comments. We have modified Figure 1A and Figure 2A by adding figures of salivary gland tissues of control mice.

Figure 1. Histopathologic changes in the submandibular glands (SMGs) of NOD/ShiLtJ mice.

Figure 2. Expression of syndecan-1 (SDC-1) in blood and submandibular gland (SMG) tissues of NOD/ShiLtJ mice.

  1. In Figure 2B, the authors measured the changes in SDC-1 concentrations in the blood, how about the IgG production levels in the blood? The relationship between IgG levels and SDC-1 levels?

[Response] We appreciate the reviewer’s critical comments.

We found that the intensity of inflammation and formation of germinal center (GC) were reduced in inflamed salivary gland tissues after HS treatment. These changes suggest that they may lead to decreased plasma cell transformation, reduced autoantibody production, and normalization of blood IgG levels, but unfortunately, blood IgG concentration and autoantibody levels were not determined in this study. In future studies, we plan to further examine the correlation between reduced GC formation and autoantibody and IgG production after HS treatment in Sjögren's disease (SD).

  1. Did the authors assess the B cell and T cell subset changes in SMG of NOD/ShiLtJ mice before and after the HS treatment?

[Response] We thank the reviewer for the comment.

We did not perform fractionation studies on blood lymphocytes in this study. In addition, salivary gland tissue was stored at -80°C after initial tissue dissection, so single cell sequencing or immunophenotyping of lymphocytes using multicolor flow cytometry could not be performed on salivary gland tissue. As an alternative, we sought to identify the subtypes of B and T cells infiltrating the salivary gland tissue by semi-quantitative RT-PCR for cell surface markers or intracellular markers.

The B cell fractions that play an important role in SD are known as marginal zone (MZ) B cells, memory B cells, and plasma cells [1]. During B cell differentiation, certain cell surface molecules are observed at different stages of differentiation, making it difficult to define a specific cell surface marker as a specific subtype of B cell by RT-PCR assay [2]. There are also cell surface markers that are expressed on cells other than B cells, such as CD1d, which is highly observed on MZ B cells but can also be observed on other antigen presenting cells. SDC-1 is predominantly expressed in plasma cells and epithelial cells in adult cells. To identify B-cell subtypes, we used CD23 and CD1d as MZ B cell markers, CD27 as memory B cell and plasma cell markers [2, 3, 4], and SDC-1 and IRF4 [5] as plasma cell markers. After HS treatment, CD27 and IRF4 were significantly reduced and SDC-1 also tended to decrease, suggesting a decrease in infiltration of plasma cells and memory B cells. Both CD23 and CD1d were significantly reduced, indicating that the MZ B cell fraction was also reduced after treatment. In conclusion, we found that all three B cell subtypes, which are important in the pathogenesis of SD, were reduced.

Next, we used cell-specific intracellular markers for subtypes of T cells, including Tbet in Th1 cells, GATA in Th2 cells, RORα in Th17 cells, and Bcl6 in follicular T helper (Tfh) cells were used to identify changes in infiltrating T cell subtypes after HS treatment. In this study, Th1 and Th2 subtypes were significantly decreased after HS treatment, but Th17 cells were increased. In addition, Tfh cells tended to decrease, but there was no statistical significance.

We have added the findings of subtype changes in B and T cells using RT-PCR after treatment with heparan sulfate (HS) to the “Results” as follows: “As single cell sequencing or immunophenotyping of lymphocytes using multicolor flow cytometry could not be performed on salivary gland tissue due to tissue conditions, we sought to identify the subtypes of B and T cells infiltrating the salivary gland tissue by semi-quantitative RT-PCR for cell surface markers or intracellular markers (Figure S2). The B cell fractions that play an important role in Sjogren’s disease (SD) are known as marginal zone (MZ) B cells, memory B cells, and plasma cells. (Witas, R.; J Clin Med 2020, 9, 3057).During B cell differentiation, certain cell surface molecules are observed at different stages of differentiation, making it difficult to define a specific cell surface marker as a specific subtype of B cell by RT-PCR assay (Edwards JCW, Nature Reviews Immunology 2006.6, 394–403.). There are also cell surface markers that are expressed on cells other than B cells, such as CD1d, which is highly observed on MZ B cells but can also be observed on other antigen presenting cells. SDC-1 is predominantly expressed in plasma cells and epithelial cells in adult cells. To identify B-cell subtypes, we used CD23 and CD1d as MZ B cell markers, CD27 as memory B cell and plasma cell markers (Edwards JCW, Nat Rev Immunol 2006.6, 394–403. / Klein U, J Exp Med 1998, 188:1679-1689/ Agematsu K, Eur J Immunol 1997, 27:2073-2079/), and SDC-1 and IRF4 as plasma cell markers (Klein U; Casola S; Nat Immunol . 2006 Jul;7(7):773-82). After HS treatment, CD27 and IRF4 were significantly reduced and SDC-1 also tended to decrease, suggesting a decrease in infiltration of plasma cells and memory B cells. Both CD23 and CD1d were significantly reduced, indicating that the MZ B cell fraction was also reduced after treatment. In conclusion, we found that all three B cell subtypes, which are important in the pathogenesis of SD, were reduced.

Next, we used cell-specific intracellular markers for subtypes of T cells, including Tbet in Th1 cells, GATA in Th2 cells, RORα in Th17 cells, and Bcl6 in follicular T helper (Tfh) cells were used to identify changes in infiltrating T cell subtypes after HS treatment. In this study, Th1 and Th2 subtypes were significantly decreased after HS treatment, but Th17 cells were increased. In addition, Tfh cells tended to decrease, but there was no statistical significance.

Figure S2. Changes of B and T cell subsets in submandibular gland tissues of NOD/ShiLtJ mice after treatment with heparan sulfate.

The semi-quantitative reverse transcription PCR experiments have been described in the “Materials and Methods” as follows: “total RNA was isolated from salivary gland tissues using TRIzol regent (Invitrogen, Carlsbad, CA), and the first strand cDNA was synthesized using Superscript III reverse transcriptase (Invitrogen). Real-time PCR was performed using a ViiA™ 7 Real-Time PCR System (Applied Biosystems, CA) using SYBR® Green Master Mix (Applied Biosystems). The oligonucleotides used are listed as follows: CXCL13: F (GGCCACGGTATTCTGGAAGC), R (ACCGACAACAGTTGAAATCACTC), CXCR5: F (GAATGACGA CAGAGGTTCCTG), R (GCCCAGGTTGGCTTCTTAT), CD27: F (AGAAGAAACCACGGGCCAAAT), R (CTCCTGGAT AGGGATAGCACTG), SDC-1: forward (F) (TCAGAGCCTTTTGGACAGGAA), reverse (R) (CACCAGGCACACAGCAAAGA), IRF4: F (AAAGGCAAGTTCCGAGAAGGG), R (CTCGACCAA TTCCTCAAAGTCA), CD23: F (ATCTCAGCCGTGATCTTGTTCT), R (TCGACAGTAGAGTAGGGTA AGGA), CD1d: F (CTGTCTGCGGGCTGTGAAAT), R (TCCCCAGAATCTCACGACATATT), CD3: F (AAGCCTGTGACCCGAGGAA), R (TGCGGATGGGCTCATAGTCT), CD4: F (AAGTGACCTTCAG TCCGGGTA), R (GGGTTAGAGACCTTAGAGTTGCT), Th1 (T-bet): F (AACCGCTTATATGTCCA CCCA), R (CTTGTTGTTGGTGAGCTTTAGC), Th2(GATA-3): F (CGGAAGAGGTGGACGTACTTTT), R (CGTAGCCCTGACGGAGTTTC), Th17(RORα) : F (CAGAGCAATGCCACCTACTCCT), R (CTG CTT CTTGGACATCCGACCA), Bcl-6: F (CCGGCACGCTAGTGATGTT), R (GCACTGTCTTATGGGCTC TAAAC) and β-actin: F(CTAAGGCCAACCGTGAAAAG), R (ACCAGAGGCATACAGGGACA). All the RT-qPCR reactions were performed in triplicate. The expression levels of interested genes were normalized to geometric mean of b-actin as internal control, and calculated by 2ΔDDCT method.”

References

  1. Witas, R.;Gupta S;Nguyen C. Contributions of Major Cell Populations to Sjögren's Syndrome. J Clin Med. 2020, 9, 3057. http://doi: 10.3390/jcm9093057
  2. Edwards, J.C.W.;Cambridge G. B-cell targeting in rheumatoid arthritis and other autoimmune diseases. Nat Rev Immunol. 2006, 6, 394–403. http://doi: 10.1038/nri1838
  3. Klein, U.; Rajewsky, K.; Küppers, R. Human immunoglobulin (Ig)M+IgD+ peripheral blood B cells expressing the CD27 cell surface antigen carry somatically mutated variable region genes: CD27 as a general marker for somatically mutated (memory) B cells. J Exp Med. 1998, 188, 1679-1689. http://doi: 10.1084/jem.188.9.1679
  4. Agematsu, K.;Yang, F.C.;Nakazawa, T.;Fukushima, K.;Ito, S.;Sugita, K.;Mori, T.; Kobata, T.;Morimoto, C.;Komiyama, A. B cell subpopulations separated by CD27 and crucial collaboration of CD27+ B cells and helper T cells in immunoglobulin production. Eur J Immunol. 1997, 27, 2073-2079. http://doi: 10.1002/eji.1830270835
  5. Klein, U.; Casola, S.; Cattoretti, G.; Shen, Q.; Lia, M.; Mo, T.; Ludwig, T.; Rajewsky, K.; Dalla-Favera, R. Transcription factor IRF4 controls plasma cell differentiation and class-switch recombination. Nat Immunol. 2006, 7, 773-782. http://doi: 10.1038/ni1357

Round 2

Reviewer 3 Report

Comments and Suggestions for Authors

Thank you for addressing my comments. Please see only my minor suggestion. Statistically significant p-values should be presented rounded to the third decimal place (in the case of very small values, p<0.001), and those with a value > 0.05 to the second decimal place.

Author Response

Thank you for addressing my comments. Please see only my minor suggestion. Statistically significant p-values should be presented rounded to the third decimal place (in the case of very small values, p<0.001), and those with a value > 0.05 to the second decimal place.

[Response] We appreciate the reviewer’s comment. I fixed the presentation of P values. In the manuscript and supplementary tables, the corrected P values are highlighted in green.

Reviewer 4 Report

Comments and Suggestions for Authors

The authors have addressed my concerns properly. I have no further questions.

Author Response

The authors have addressed my concerns properly. I have no further questions.

[Response] We thank the reviewer.